# Evolution of Cellular Immunity Effector Cells; Perspective on Cytotoxic and Phagocytic Cellular Lineages

**DOI:** 10.3390/cells10081853

**Published:** 2021-07-22

**Authors:** Edna Ayerim Mandujano-Tinoco, Eliya Sultan, Aner Ottolenghi, Orly Gershoni-Yahalom, Benyamin Rosental

**Affiliations:** 1The Shraga Segal Department of Microbiology, Immunology, and Genetics, Faculty of Health Sciences, and Regenerative Medicine and Stem Cell Research Center, Ben Gurion University of the Negev, Beer Sheva 8410501, Israel; eliyasul@post.bgu.ac.il (E.S.); anero@post.bgu.ac.il (A.O.); orlyge@post.bgu.ac.il (O.G.-Y.); 2Laboratory of Connective Tissue, Centro Nacional de Investigación y Atención de Quemados, Instituto Nacional de Rehabilitación “Luis Guillermo Ibarra Ibarra”, Calzada Mexico-Xochimilco No. 289, Col. Arenal de Guadalupe, Tlalpan, Mexico City 14389, Mexico

**Keywords:** innate immunity, adaptive immunity, phagocytosis, cytotoxicity, comparative immunology

## Abstract

The immune system has evolved to protect organisms from infections caused by bacteria, viruses, and parasitic pathogens. In addition, it provides regenerative capacities, tissue maintenance, and self/non-self recognition of foreign tissues. Phagocytosis and cytotoxicity are two prominent cellular immune activities positioned at the base of immune effector function in mammals. Although these immune mechanisms have diversified into a wide heterogeneous repertoire of effector cells, it appears that they share some common cellular and molecular features in all animals, but also some interesting convergent mechanisms. In this review, we will explore the current knowledge about the evolution of phagocytic and cytotoxic immune lineages against pathogens, in the clearance of damaged cells, for regeneration, for histocompatibility recognition, and in killing virally infected cells. To this end, we give different immune examples of multicellular organism models, ranging from the roots of bilateral organisms to chordate invertebrates, comparing to vertebrates’ lineages. In this review, we compare cellular lineage homologies at the cellular and molecular levels. We aim to highlight and discuss the diverse function plasticity within the evolved immune effector cells, and even suggest the costs and benefits that it may imply for organisms with the meaning of greater defense against pathogens but less ability to regenerate damaged tissues and organs.

## 1. Introduction

Over hundreds of millions of years, the development of defense mechanisms in multicellular organisms to fight the invasion of microbes and viruses has represented an essential cue for the evolution of life. Not surprisingly, immune systems have diversified into a highly complex set of cells and molecular mechanisms to target pathogens and also to allow regeneration, tissue repair and the recognition of foreign transplants.

Evolution of animal immune systems can be analyzed during a time span of at least 1000 million years. From the most primitive multicellular organisms (i.e., sponges) to protostomes organisms (i.e., Platyhelminthes, mollusks, nematodes and arthropods) to chordate invertebrates (i.e., Tunicata) defense and self/non-self-recognition responses rely completely on innate immunity. Jawless vertebrates such as hagfish and lampreys were the first to connect innate responses with some adaptive elements. Whereas all jawed vertebrates ranging from primitive sharks to mammals are equipped with innate and adaptive immunity [1,2].

In general, innate immunity is a fast and non-specific response associated with the presence of humoral and cellular elements [3]. By contrast, adaptive immunity uses the induction of specialized cells such as B and T lymphocytes and molecules including the major histocompatibility complex (MHC), B-cell receptors (BCR), T-cell receptors (TCR), immunoglobulins (Ig), and antibodies to confer immunological memory and very high specificity thus fighting against a previously recognized infection [4]. Both kinds of immune responses rely on two main cellular activities which are phagocytosis and cytotoxicity. These cellular immune mechanisms have been found at the earliest evolutionary stages of multicellular animals and diversified into a wide heterogeneous repertoire of effector cells through evolution.

Here, we will review how phagocytic and cytotoxic effector cell lineages have evolved against pathogens, in histocompatibility recognition and in killing virally infected cells, highlighting the common molecular features shared between the more evolved species with their early invertebrate ancestors. We emphasize on multicellular organisms, where we assume existence of specialized immune effector cells subpopulations. Finally, we aim to discuss the functional plasticity of these immune cells in the clearance of damaged cells, and their repercussions on regeneration and tissue repair capacities.

## 2. Phagocytosis in the Evolution of Immune Response

Phagocytosis is the process by which particles (>0.5 μm) are recognized, bound to a plasma-membrane envelope, and internalized into an organelle called “phagosome”. Two major classes of particles are engulfed by phagocytes: foreign microorganisms and “altered self” (apoptotic and necrotic cells) particles. Élie Metchnikoff was the first to realize the significance of phagocytosis for cellular immunity in processes such as the host response to injury and infection, inflammation, and tissue homeostasis [5,6,7]. Phagocytosis is highly conserved between species and has essential roles for the development and maintenance of multicellular organisms. The phagocytic process requires a coordinated sequence of events that begins with the engagement of plasma-membrane receptors, which allow the recognition of damaged or infected cells and different classes of pathogens. This leads to the activation of signaling pathways needed for the rearrangement of the cytoskeleton and the internalization of the particle. Once internalized, the phagosome dynamically undergoes fusion and fission events with intracellular secretory vesicles to finally mature into a phagolysosome. The hydrolytic activity of phagolysosomes results in the digestion of the internalized cargo [5].

In invertebrates, “professional phagocytes” have a macrophage-like phenotype and express receptors that recognize pathogen-associated molecules. Ligand-receptor binding induces different signaling cascades leading to the production of effector molecules to eliminate or inactivate the infection. These include oxygen-dependent (superoxide anion, hydrogen peroxide, hydroxyl radical) and oxygen-independent (lysozyme and defensins) molecules [8]. Phagocytic cell lineages and mechanisms to engulf and combat microbes and viruses have been diversified in vertebrates, but have several homologous molecules and cells with their lower ancestors [2,8].

### 2.1. Phagocytic Effector Cells

Amebocytes are the most primitive animal cells with phagocytic ability, these are found in sponges and cnidarians acting in the recognition of nutrients and foreign elements [9]. We have recently characterized two populations of phagocytes (granular spheroid and ameboid phenotype) from members of the Hexacorallia class, the coral *Pocillopora damicornis* and the sea anemone *Nematostella vectensis*, finding that they engulf bacteria, fungal antigens, carboxylated beads and self-damaged cells. We showed that this phagocytosis is different than pinocytosis which is performed by the majority of the cells [10]. Reticular cells, the phagocytic cells of free living planarians from the phylum Platyhelminthes, can migrate towards heat-shocked bacteria and phagocytose them. Their pseudopodia and the modes of their endocytosis can be morphologically compared to human neutrophils [11,12]. From the phylum Annelida, the earthworm *Eisenia andrei* has several subtypes of coelomocytes including eleocytes and granular amoebocytes [13]. In arthropods, several cell types have been characterized. For example, *Drosophila melanogaster* has hemolymph circulating cells collectively called hemocytes. Two types of hemocytes have been identified in the larval stage: Plasmatocytes, which phagocytose microorganisms and cell debris and secrete signaling molecules (i.e., Eiger, Upds, drosocin, defensins) with high similarity to mammalian macrophages and neutrophils, according to transcriptional profiles; and crystal cells, which activate the melanization cascade upon wounds or infections [14]. However, single-cell RNA sequencing studies showed that these cell types are more heterogeneous as 14 different functional clusters of hematocytes have been recently discovered, including lamellocytes, which are activated immune cells that only differentiate upon immune induction [15]. The tunicate *Botryllus schlosseri* has a myeloid lineage of phagocytic cells with high similarities and gene set homology to mammalian myeloid lineage [16], and also amoebocytes and large phagocytes suggested to be related to arthropods and echinoderms [17]. In bony fishes, the phagocytic armament includes macrophages, monocytes, dendritic cells, neutrophils, granulocytes, eosinophils, basophils and mast cells [18]. While in mammals, these professional phagocytes are able to differentiate into highly specialized subtypes of cells which exert different cytokine profiles, exclusive functions in regeneration and infection fighting (i.e., M1 and M2 macrophages), tissue specificity and also novel molecular mechanisms such as the antibody-dependent cellular phagocytosis (ADCP) for the clearance of microorganisms and immune complexes [19,20].

### 2.2. Receptors

Phagocytic cell lineages possess cell-surface receptors as the mechanism to respond and initiate the phagocytic process and the production of killing molecules. Primitive sponges possess LPS binding receptors and intracellular receptors “Nucleotide-binding domain and Leucine-rich Repeat” (NLRs) to bind fungal polysaccharides, bacteria and virus [21,22]. Toll-like receptors (TLRs) are ancient sensors and the best characterized in detecting and respond against invading pathogens. TLRs have been traced to several invertebrate species as their emergence predate the separation of bilaterians and cnidarians. In *D. melanogaster*, Toll receptors have a crucial role in immune defense. Flies deficient in Toll members are severely immunocompromised to response against fungal and bacterial infections [23]. Moreover, the Down syndrome cell-adhesion molecule (DsCAM) acts as a phagocytic cell-surface receptor that binds to bacteria such as *Escherichia coli* [24]. A new study in *B. schlosseri* has identified BsTLR1 as a member of the TLR family which is actively transcribed in phagocytes and morula cells as a mechanism of non-self recognition [25]. Interestingly, TLRs are highly diversified in urchins (253 genome sequences), more than in vertebrates [26]. Agnathans have seven identified TLRs, while in mammals this number is up to 13, which have high specificity in the ligands they recognize, suggesting a host–ligand coevolution [27]. Although TLRs have not been identified in planarians, extracellular leucine rich repeat (LRR)-motifs are coded in their genome [28]; however, the exact role LRRs play in the immune response is unknow. Other types of receptors are highly conserved through evolution. For example, the scavenger receptors Croquemort (Crq) and Peste of *D. melanogaster* have convergently evolved with their mammalian orthologues CD36 and SR-BI, respectively [29]. There are also three predicted CD36-family homologs in the genome of *Caenorhabditis elegans* [30]. They are CD36-like family members and major plasmatocyte markers that recognize apoptotic cells and bind to surface lipopetides of different Gram-negative and Gram-positive bacteria and fungi. Both are required to efficiently eliminate infections, and their absence causes poor phagocytic plasmatocytes and defects on phagosome maturation [31]. The Transformer (Trf) proteins also have important roles in the immunity of sea urchins by inducing the phagocytic activity of coelomocytes to directly engulf bacteria in an ADCP resembling matter [32,33]. SpTrf genes have two exons, recombination mechanisms between the gene elements of the second exon result in the fact that sea urchins express more than 260 different Trfs proteins, single one recombinant protein expressed in an individual cell, which act synergistically to detect a variety of pathogens thus providing an efficient immunological army [34,35]. Furthermore, no Trfs have been identified outside the echinoid lineage [36], indicating that there are several unique immune mechanisms that should be studied in sea urchins. Moreover, this discovery suggests that there are diverse mechanisms of immune receptor diversification, which are different from what we know in vertebrates.

### 2.3. Effector Molecules

After the accumulation of phagocytes in a localized area of physical and chemical insult, these release several molecules to allow recognition, pathogen killing, and inflammation. In invertebrates these molecules include oxidative killing (ROS and NOS species), agglutination, clotting, coagulation (i.e., fibrinogen-related peptides FREPS), melanization (cytotoxic quinones, melanin) and antimicrobial peptides. In the hexacorallian *P. damicornis* and *N. vectensis*, a high production of ROS is associated with phagocytic activity [10]. In the protochordate *B. schlosseri* the rhamnose-binding lectin (BsRBL) activate phagocytes and induces the synthesis and release of cytokines recognized by the antibodies anti-IL1α and anti-TNFα [37]. Mammalian phagocytes secrete a wide array of chemokines (CXC) and cytokines. Cytokines have been divided into different classes: interferons (IFNs), interleukins (ILs), tumor necrosis factors (TNFs) and transforming growth factors (TGFs). Studies have been conducted to search for homologs in lower species. Some worms and mollusks have shown to enhance phagocytosis in response to the cytokines TNF-α and IFN-γ, but no encoded genes have been characterized to explain their function [2]. *Drosophila* genome encodes different cytokine types with certain mammalian homology: Eiger is homolog to TNF, Upds is similar to type I cytokines and Spätzle is related to IL-17F. Flies with mutations in these genes are highly sensitive to bacterial infections and have less capacity to remove dead cells, showing its important role in innate immunity [38]. Transforming growth factor β (TGF-β) modulates inflammation and repair functions. The TGF-β family member, tgfβ-f, has been identified in *B. schlosseri* although its principal knowing function is related with germ cell differentiation [39]. Genes of all the major mammalian cytokines and CXC have been identified in fishes. These include pro-inflammatory cytokines IL-1 β, TNF-α and IL-6 and cytokines associated with the adaptive immunity such as IL-2, IFN-γ, IL-4/13 (homologue to IL-4 and IL13), IL-10, IL-17A/F, IL-21, IL-22 and TGF-β family members, providing strong protection against diverse viral infections (i.e., haemorrhaic virus) [40]. Although cytokines are highly conserved among vertebrates, distinct differences have been identified between classes. For example, IFN-γ is the only type II IFN identified in mammals, whereas multiple type II IFNs have been found in lower vertebrates. In contrast, type III IFNs appear to be unique to mammals. IL-1, IL-2 and IL-8 are the most conserved interleukins among vertebrates, while IL-4, IL-5, IL-10, IL-13, IL-20, IL-21 and others have been only characterized in mice and humans with very specific biological functions [41].

### 2.4. Signaling Pathways

The coordination of the phagocytic process in an integrated network of signaling cascades must have occurred since the most primitive forms of life (Figure 1). We have stated that TLRs are found in phagosomes of many invertebrates. The Toll pathway activates the nuclear factor-κB (NFκB) signaling, a master regulator of the phagolysosomal degradation of pathogens. Genome and transcriptome analysis have shown that homologs of mammalian NFκB transcription factor, its inhibitors (IκB) and activators are traced to sponges. In the demosponge *Amphimedon queenslandica* the Aq-NF-κB has been identified, this is structurally similar to NFκB p100/p105 among vertebrate subfamily transcription factors with DNA binding domains and a C-terminal that allows its translocation to the nucleus [42]. Homolog transcripts of upstream NFκB pathway components have been also identified in cnidarians. In *Drosophila*, Dorsal/DIF and Cactus has been described as homologs to NFκB and IκB, and several NFκB binding motifs were identified in promoters of immunoresponsive genes of phagocytic cells, suggesting its conserved role in the immune response (Figure 1) [43]. Furthermore, phagocytes of *B. schlosseri* activate Ras-like small GTPases, MAPKs and NFκB signaling networks as a response to the recognition of foreign cells (Figure 1) [44]. In mammalian macrophages, NFκB signaling is highly dynamic, specific, and strictly regulated acting in direct and indirect crosstalk with many other signaling pathways during an immune response, depending on the stimuli given by each interacting pathogen. This complexity allows macrophages and several other cell types to have diverse mechanisms of action thus mounting an effective immune response and in parallel protecting the tissues of possible damages caused by hyperinflammation (Figure 1) [45].

## 3. Cellular Cytotoxicity in the Evolution of Immune Response against Abnormal Self and Pathogens

Cytotoxicity (cell-mediated killing) is the process in which certain cells secrete “toxins” that lysate and neutralize unwanted target cells (i.e., pathogens, viral-infected cells, tumoral cells, foreign transplanted cells). Cooper has defined that through evolution, immune-mediated cytotoxicity could be classified into level I and level II cytotoxicity. Level I in mammals is mediated by macrophages and neutrophils which release toxic granules in the inflammatory area. They are considered the most basic manifestations of cytotoxicity and are present in invertebrates (sponges, coelenterates, sipunculids, annelids, mollusks, arthropods, echinoderms and protochordates). In contrast, level II or acquired cytotoxicity is mediated by cells that require a specific induction to be activated, through specific recognition, such as allogeneic responses, or induced by adaptive molecules. These include cytotoxic T lymphocytes (CTL), natural killer (NK), and antibody-dependent cell-mediated cytotoxicity (ADCC). It seems that these cells appear later in evolution [2,48], but the characterization of specific allogeneic responses suggests that they might be in more animal groups than what is currently known. Since the work of Cooper in 1980, NK cells were shown to have specificity and recognition of targets for cytotoxicity trough immunological synapse without the killing of adjacent cells [49,50], for that reason we moved NK-like recognition based cytotoxicity mechanism to level II cytotoxicity, compared to the original definition by Cooper [48].

All cytotoxic cells share certain characteristics such as (1) the ability to trigger cell death of target cells, (2) the presence of adhesion molecules allowing their adhesion to the target, (3) their activity is mediated by monokines and lymphokines [51].

### 3.1. Effector Cells

All animals have several immune cells with cytotoxic activity to lyse host and foreign cells. Regarding level I cytotoxicity, those cells have been found in different invertebrate organisms as probably the most ancient type of cytotoxic immune response. For example, killer cells from sipunculid worms exert cytotoxic effects on allogeneic erythrocytes, but not on erythrocytes from worms that reside closely [52]. In the freshwater pulmonated mollusk *Planorbarius corneus*, a class of hemocyte that has been morphologically characterized as round hemocytes (RH) exerts cytotoxic activity on the human erythroleukemia K562 cells thus participating in the graft rejection of allo- and xeno-grafts [51]. In arthropods, the amoeboid hemocytes of *Limolus polyphemus* are stimulated by bacterial lipopolysaccharide (LPS) endotoxins to secrete great amounts of clottable protein and to produce nitric oxide (NO) synthesis as a cytotoxic mechanism to protect the host from invading pathogens [53]. Additionally, in in vitro experiments, crayfish (*Astacus astacus*) granular and semi granular hemocytes with phenoloxidase and laccase activity display cytotoxic effects towards different mammalian tumor and non-tumor cells [54,55]. Coelomocytes from the sea urchin *Arbacia punctulata* exert cytotoxic activity against human and murine target cells. A particular population of these phagocytic cells is the one with the highest cytotoxic activity which are positive to the human NK markers CD14, CD56 and CD158b [56]. These cytotoxic cells have also found in *Strongylocentrotus droebachiensis, S. pallidus* and *Echinus esculentus* in which targeted-cell death is characterized by cell detachment, disintegration and the formation of a multinuclear non-cellular protoplasm [57].

At this point we do not think that phagocytosis of damaged self-cells should be considered as level 1 cytotoxicity. Although, we do think that where phagocytosis of damaged self-cells is found, it can be suggestive of potential level 1 cytotoxicity (as the mechanisms of altered-self recognition exist) and should be further investigated (Figure 2). For instance, the recent discovery of phagocytosis of heat-stressed cells in ex-vivo experiments in Hexacorallians [10].

For level II of specific cytotoxicity, in the tunicate *B. schlosseri*, morula cells (MC) have been characterized as cytotoxic cells. They contain phenoloxidase, accumulate in rejection points and morphologically resembles NK cells. While 15% of their genes are shared with cytotoxic lymphocytes of vertebrates, MC express 85% tunicate-specific gene repertoire [16,17], suggestive of convergent lineage more resembling other invertebrates phenoloxidase based immune cells (Figure 2). The cytotoxic activity of MCs is involved in a natural process of allorecognition that occurs when two *B. schlosseri* colonies with different genotypes touch [58]. This self–nonself recognition is regulated by a single polymorphic histocompatibility gene: the Botryllus Histocompatibility Factor (BHF) [59]. The sharing of at least one BHF allele is a determinant for the fusion or rejection of the colonies [16,17]. Interestingly, MC express high levels of the polymorphic gene FuHC which could be a base for BHF recognition mechanism [17,60] (Figure 2).

Transcriptome analysis and cellular characterization studies in the jawless vertebrates, lampreys, have proposed that two T-like cell populations expressing the variable lymphocyte receptors (VLRs) VLRA and VLRC, respectively, carries cytotoxic activity [61,62]. These cells express several genes including cell surface receptors (i.e., CD45, TCR, VpreB, paired-Ig-like receptors), cytokines (IL17) and transcription factors (i.e., GATA 2/3, c-Rel, BCL11b) resembling those that T lymphocytes use to migrate, proliferate and differentiate in jawed vertebrates [63]. Lymphocyte lineages have also been found in the hagfish, however these have not been fully characterized [61,64]. This shows a whole convergent parallel adaptive immune system of lymphocytes, which is based on LLRs instead of Ig superfamily like the jawed vertebrates (Figure 2).

In jawed vertebrates, the cytotoxic cells NK and CTLs belong to the lymphoid lineage. NK cells do not express antigen-specific receptors but are highly capable to recognize and kill tumor and viral-infected cells [40]. NK cell function has been recognized in representatives of all vertebrate classes [65]. Comparative studies using high-quality genome assemblies have shown that NK cells diversity in mammals has a big influence on species-specific immune responses and outcomes of pathogen infection [66]. NKT-like cells have been identified in fishes [65,67] the frog *Xenopus laevis* [68] and through the identification of CD1 molecules, it is suggested that this kind of cells are also present in reptiles and birds [65].

CTLs have three important elements to recognize antigens expressed by viral-infected cells or cancer tissue: (1) a diverse repertoire of Ig domain-based receptors, (2) T cell receptor (TCR) and (3) major histocompatibility complex (MHC). Genes of these elements and the presence of CTLs is a common feature found in all jawed vertebrates which include cartilaginous and bony fishes, amphibians, reptiles, birds, and mammals. Moreover, these cells were also found in the extinct placoderms [65,69]. In mammals, also activated macrophages could selectively lyse malignant cells in a contact-dependent and non-phagocytic process, having important roles in the restoration of tissue homeostasis [70]. Finally, ADCC is well characterized in mammals by eosinophilic and neutrophilic granulocytes, NK cells, and CD16^+^ blood monocytes [71]. Their main effects are associated with graft rejection, autoimmune diseases, tumor surveillance, antiviral and antiparasitic defense [71,72,73,74].

We think that in cases where allogeneic specificity is found in cytotoxicity, this should be considered and further investigated as potential for level II cytotoxicity. For instance, the example mentioned earlier of sipunculid worms showing different cytotoxic effects between levels of allogeneic proximity of the target erythrocytes is suggestive of specificity recognition by the cytotoxic effector cells [52]. Characterization of the cells and those recognition mechanisms would shed light if there were a presence of level II specific cytotoxicity (Figure 2). Moreover, receptor recombination as an example of Trfs in urchins could also suggest specific cytotoxicity, at this stage it was only tested against bacterial opsonization [32,33], but if it is effective against foreign cells or abnormal self, then could be considered as level II specific lysis.

### 3.2. Receptors

Killer cells possess diverse receptors for triggering their cytotoxic functions, these receptors seem to appear early in evolution although without high specificity. In mollusks, the Fibrinogen-related proteins (FREPs) bind to parasites and has been suggested as potential adaptive defense molecules [75,76,77]. Hemocyte membranes of the arthropod *P. bicarinatus* have trypsin-labile receptors and nonspecific cytotoxic cell receptors protein 1 (NCCRP-1) which appear to be mediators of allorecognition [78]. Additionally, haemocytes from *Prodenia eridania* express non-specific receptors for foreign particles [48]. NK cells from mammals have Killer Inhibitory Receptors (KIRs) which inhibit the activation of cytotoxic programs when recognize self-MHC, according to the “missing self” hypothesis NK cells kill in this manner allogeneic foreign cells that do not express self-MHC or that have altered expression of MHC molecules [79,80,81]. In humans, CD94, NKG2 and NKR-P1 are examples of these receptors, while rodents express a group of Ly49 receptors. Blood cells of the tunicate *B. schlosseri* also express a gene coding for a type II transmembrane protein with high similarity to vertebrates CD94 and NKR-P1, which is upregulated during allorecognition [82]. VLRA, VLRB and VLRC genes have been identified in lampreys and hagfish [61,62]. These receptors are highly diverse due to a somatic diversification strategy based on the insertion of LRR sequences in a single germline gene [83]. The locus VLRA has >500 LRR cassettes, while there are >800 LRR cassettes in the VLRB locus and >200 LRR cassettes in the VLRC locus [84,85]. This evolutionary mechanism to diversify lymphocyte receptors at a single cell level is similar, but not identical, to the multigene recombinatorial strategy of immunoglobulin gene segments and TCR molecules in jawed vertebrates [83,86].

In all jawed vertebrates, CTLs recognize peptides bound to MHC molecules by expressing the T cell receptor complex (TCR). Peptides are derived from viral or bacterial proteins during infection [87], and MHC I nonself recognition also occurs after allotransplantation [88]. In this sense, the expression of TCRs is clonotypic since one T cell expresses a single TCR specificity that is determined by both the antigen peptide and the MHC determinants. TCR diversity, which is generated by mechanisms of genomic rearrangement, and the fact that MHC molecules are highly polymorphic allow organisms to counteract pathogen evasion of T cell responses [89]. While TCR is the major CTL marker, the glycoproteins CD3, CD8 and CD57 are other important markers which bind MHC molecules [89]. The structure and function of immunoglobulin M (IgM), the classic TCRs (αβ chains) and MHC class II are conserved in jawed vertebrates from cartilaginous and bony fishes, to amphibians, reptiles, birds, and mammals. However, other immune receptors as γδ TCR, NKRs and nonclassical MHCs are present but with diverse function, which in most cases is still unexplored [65,90,91,92].

### 3.3. Effector Molecules

As in phagocytosis, the secretion of chemokines and effector molecules is important for the activation of cytotoxic programs which are an important immune army defense against pathogens and foreign transplanted cells. Between the most conserved cytotoxic components among invertebrate humoral defenses is lysozyme. This is an enzyme that digest components of the cell-wall from Gram-positive bacteria. It has been found in the soluble fraction of hemocytes preparations from mollusk, freshwater crayfish, freshwater snail, earthworms and ascidian. Two types of lysozymes are also found among vertebrate animals that are the Chicken (c)-type lysozyme and the goose (g)-type lysozyme [58]. Other soluble factors with antibacterial/antiviral activity (proteinases, pore forming in cell membranes or ion gradient disruptors) including astacidines, hemagglutinin and anti-LPS factor have been identified in the hemolymph of many invertebrates [93]. The production of reactive oxygen intermediates (O_2_, H_2_O_2_, HOCl) as well as reactive nitrogen intermediates (NO, N_2_O_3_, ONOO, NO_2_, NO_3_) has been found associated with cytotoxic activity in echinoderms, coelenterates, nematodes, annelids, insects, crustaceans and mollusks [94]. The production of Nitric Oxide (NO) has been detected in two parasitic and two free living flatworms, in these organisms the presence of phenoloxidase has also been detected although its function in their cytotoxic immune responses is still unknown [12]. In mammals, ROS/NOS-mediated cytotoxicity by macrophages, NK cells and monocytes has also been associated with the eradication of pathogens, viral infections, and antitumor response. Additionally, it was also associated with the modulation of inflammation during tissue repair and wound healing processes in vertebrate species including humans and rodents [95,96]. Vertebrate activated macrophages contain primary lysosomes with more than 40 enzymes with proteolytic activity including acid hydrolases, peroxidases and neutral proteases [97]. The most important cytokines produced by human NK cells are TNF-α, IFN-γ and IL-22. These promote the killing of intracellular pathogens and the skewing of T adaptive immune responses [98]. NK-like from mollusks also contain TNF-α, however its effects on cytotoxic killing remains to be studied [99]. Upon MHC recognition, CTLs generate typical lysosomal granules with hydrolytic and pore forming proteins (i.e., acid phosphatase, β-glucoronidase, acid esterase, serine proteases and perforin) in order to lyse their target cells [100].

### 3.4. Signaling Pathways

After the engagement of cytotoxic cell receptors that recognize target cells, phosphorylation cascades result in the activation of key signaling pathways for the execution of cytotoxic programs including the release of lysosomal granules. In NK cells, this signaling is primarily mediated by Src family kinases, MAPK activation mediated by PI3K and Rac-1 and ERK pathway. In ADCC, a Ras-dependent pathway also guides to ERK activation and thus to granule releasing. CTLs classically use the TCR signaling pathway triggered by ligation of peptide-MHC. This involves the recruitment of Ras, Raf-1 activation, phosphorylation of MEK-1 and ERK phosphorylation and activation [101,102]. Evidence suggests that the cytotoxicity of molluscan and *Drosophila* hemocytes relies in signal transduction pathways that are similar to vertebrate response including protein kinases A and C, focal adhesion kinases, Src, FAK and MAPK proteins, small G proteins (Ras), and second messengers as cAMP and Ca^2+^ [103]. In the protochordate *B. schlosseri*, immune signal transduction pathways include the activation of G-proteins, protein kinases A and C, PI3K and MAPKs, although these have been more related with phagocytic than with cytotoxic programs [104].

## 4. Perspective on the Effect of the Immune System in the Evolution of Regenerative Capacities

Regeneration is the capacity of tissues and organs to reconstitute their structure and function after injury. Regenerative potential has been lost through the evolution of animals. Basal invertebrates have extraordinary abilities to regenerate complex structures and even an entire organism. Among vertebrates, fishes and amphibians have a well-developed capacity to regenerate tail, jaws, spinal cord, lens, myocardium, and some gastrointestinal regions. Regeneration of complex structures in mammals is limited to antlers, earholes, nipples, digit tips and liver, instead they have remarkable capacities to renew and repair tissues and healing wounds (i.e., skin, bone, skeletal muscle, hematopoietic system) [121,122].

Several questions about the evolution of the regenerative potential have been discussed. Is it a normal development epiphenomenon that has been selectively lost? Has it lacked selective pressure to be preserved in vertebrate taxa as an adaptive trait? What physiological benefits predominate over the ability to regenerate during the evolution of life?

The immune system has pivotal roles in both antimicrobial/antiviral response, and tissue repair and regeneration. At the wound site, phagocytic and cytotoxic immune cells (i.e., neutrophils, macrophages, monocytes, lymphocytes) are recruited to clear debris and to secrete signaling molecules activating inflammation, proliferation, differentiation and extracellular matrix remodeling programs [123]. In this regard, the group of Anthony Mescher proposed in 2003 an immunologic hypothesis stating that the appearance of novel cell types (and their interactions) during the evolution of the adaptive immune system allowed vertebrates to have better response against pathogens in injured tissue but, these advantages implied the possible loosing of regenerative capacities. This has been reviewed in detailed in [124,125].

Mescher’s hypothesis relies on experimental studies in different models of regeneration: full-thickness injuries in fetal skin [126], transgenic mouse models deficient for specific immune cells [127], hind-limb amputation in urodeles and anurans [128]. The given evidence suggests that the profile of the infiltrated immune cells and their interactions within the injured tissue directly influences inflammation and scarring [125,129,130].

It has been proposed that inflammation and scarring prevent tissue/organ regeneration. This is supported by evidence showing that vertebrates with less well-developed adaptive immune system (such as teleost fish, larval anurans, urodele amphibians) show minimal inflammatory responses to injury and excellent regenerative potential [129]. Furthermore, the success of the regenerative process is also dependent on the timely resolution of inflammation, which requires changes in the macrophage populations and in the gene expression of other immune cells. The regeneration of larval *Xenopus* hindlimbs is reduced when this process is anticipated or delayed [131]. Moreover, scar-free repair of fetal skin wounds has been reported in diverse mammals. In humans, this occurs until the early second trimester of gestation, after which there is a switch to the scarring repair of the adult skin healing. A remarkable difference between both repairing processes relies in the development of an immune system that causes inflammation in wounded tissues, while in fetus there is a minimal acute inflammatory response which is quickly resolved [125]. In this sense, the genetic ablation of macrophages and neutrophils in the paw of PU.1 null mouse resulted in a scarless healing process [132,133]; while wounded fetal skin of IL-10 deficient mice (anti-inflammatory cytokine) form scar and increase the inflammatory infiltrate [134]. This regenerative potential in fetuses is, however, dependent on the type of injury since severe skin wounds produce high migration of macrophages, local cellular necrosis and scar formation [135,136]. Scarring is strikingly linked to inflammation; it is caused by immune cell interactions that produce several cytokines that activate mesenchymal cells and fibroblasts, promoting fibroplasia, contraction, and collagen production. According to the data, the strength of the inflammatory response correlates strongly with the amount of the scar that the tissue or organ will form, for more information review [137].

However, research in different regenerative models has shown that specific inflammatory components (cells, cytokines and chemokines) are central promoters of regeneration. This has been deeply reviewed in [138]. In the mice ear-hole model, during the early acute inflammatory response, the overexpression of inflammatory genes related to progenitor mast cells (i.e., MCP-7, CD34, CD14/CDw17) promotes the ear-hole closure [139,140]. A strong infiltration of leukocytes occurs during the skin regeneration in adult axolotls [141], and macrophage deletion reduces zebrafish embryonic and adult tail fin regeneration, which does not occur in their caudal fin [142,143,144]. The use of glucocorticoids to reduce inflammation at the moment of amputation can fully prevent regeneration of *Xenopus* limbs [145] evidencing that regeneration does not occur without inflammation.

All this evidence shows that inflammation is essential for triggering the process of regeneration; however, the timely resolution of the inflammatory phase will also determine the success of the generation of new and functional organs or tissues. More research into immune activities during various biological responses is needed to understand how they have evolved simultaneously in such a way in which the specialization of some responses (against infections) may have negative effects on capabilities that appear to be advantageous for life, such as regeneration.

## 5. Concluding Remarks

Researchers are still dealing with the understanding about the origin and phylogenetic evolution of both phagocytosis and cytotoxicity functions in the immune response. More integral analysis with evolutive perspectives are needed to provide a broad overview about the commonalities and divergences between immune systems of different organisms. For this, we require to establish simple animal models that allow the study of complex responses in the defense of an organism against a pathogen, in the recognition of foreign tissues or in the regeneration or repair of a damaged organ/tissue. With the available information, it is easier to speculate that complex immune system responses arose in ancestral organisms more than 550 million years ago, and that selective pressures influenced the diversification of effector cells and molecules but conserving essential activities as the basis to maintain homeostasis and life.

Beyond the basic research of the evolution and diversification of the immune system, there is a potential for discoveries which could be relevant to biomedical research and applications. Some very interesting and important discoveries came from comparative immunology research, such as: (I) the first discovery of immune “professional” cells and phagocytosis was done in starfish larvae [7]; (II) the antiviral mechanism of siRNA was elucidated in *C. elegance* [146], which led to the discovery of other silencing mechanisms conducted by microRNAs [147] and the widespread use of synthetic siRNAs for RNA interference in mammalian cells [148]; (III) Toll and TLR immune receptors were first discovered in *D. melanogaster* [149,150]; (IV) a new and convergent adaptive immunity, based on leucin reach repeats (LLR) was discovered in jawless vertebrates [83]; (V) stem cell competition in transplantation was first revealed in Tunicates [151], and later shown in mammalian development, aging, cancer and atherosclerosis (reviewed in [152]); (VI) CRISPR-Cas was discovered as an anti-phage mechanism in bacteria [153,154]. It is important to state that four of those discoveries (I–III, VI) were awarded the Nobel Prize due to their significance in biomedical research and medical applications [155,156,157,158]. Those examples show the potential of discoveries which could be revealed trough comparative immunology approach.

## Figures and Tables

**Figure 1 cells-10-01853-f001:**
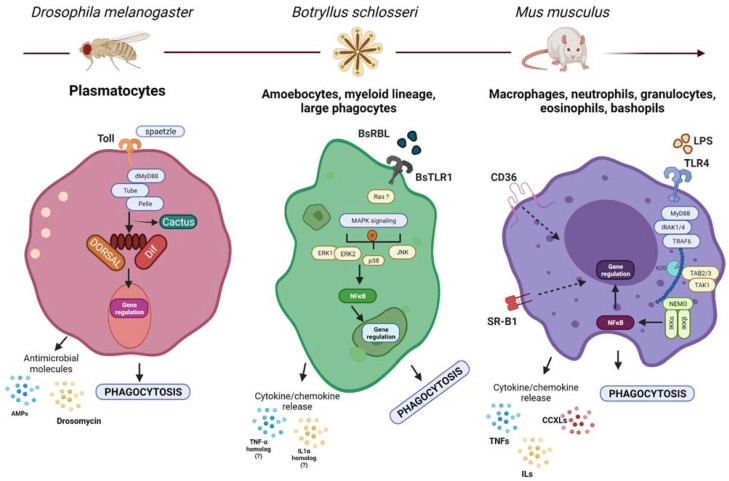
Molecular mechanisms guiding phagocytosis in different organisms. Although the phagocytic lineages and their molecular mechanisms guiding the process of phagocytosis could be different, they share several basic mechanisms with their ancestral form. Shown here: (**left**) the Toll pathway in *Drosophila* is activated by spaetzle, leading to an intracellular signaling that results in the degradation of cactus and the concomitant release of DIF/DORSAL which translocate to the nucleus and activate the transcription of immune genes including AMPs and drosomycin, which secreted as a response to infection [46]. (**center**) BsTLR1, Ras, MAPK, NFκB and cytokine homologs have been associated with the modulation of phagocytosis in *B. Schlosseri.* (?)—Represents missing functional or molecular validation. Similarly (**right**), mammalian TLRs are activated by microbial- and self-derived products leading to the activation of intracellular signaling cascade that results in the translocation of NFκB to the nucleus to activate the transcription of different cytokines and chemokines which enhance the phagocytic activity of the cells [47]. Created with BioRender.com.

**Figure 2 cells-10-01853-f002:**
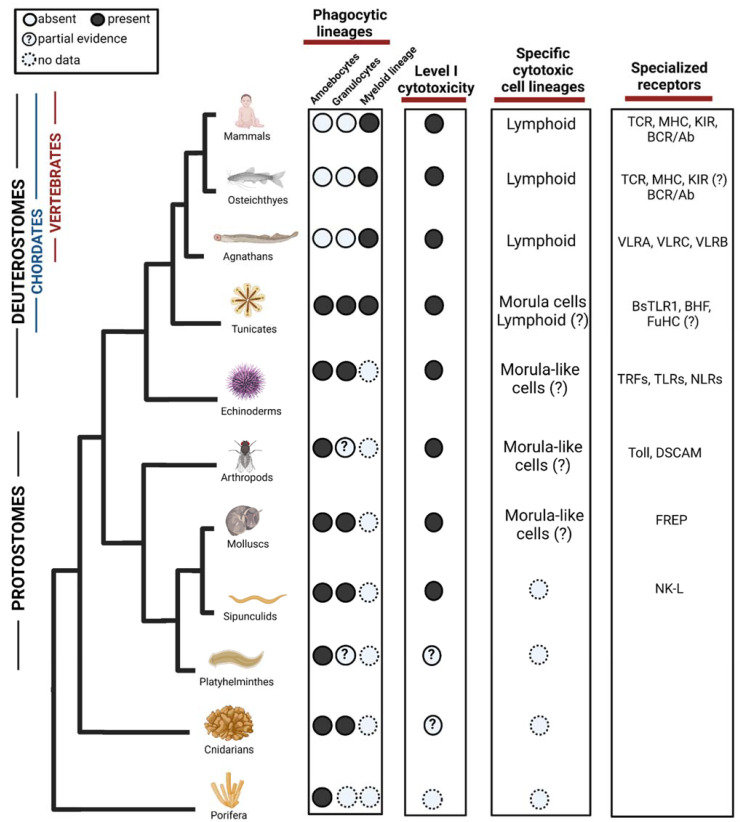
Evolution of phagocytic and cytotoxic cell lineages in the immune system. Among phagocytic lineages, amoebocytes and granulocytes are present in invertebrates, while myeloid lineage seems to evolve in the ancestor of tunicates and vertebrates. Cells representing the Level I cytotoxicity exist in Protostomes but might appear in cnidarian. Among specific cytotoxic cell lineages, morula cells have been found in tunicates although partial evidence suggests that they may evolve in earlier ancestors. Lymphoid cells are found in agnathans and have diversified in vertebrates, there are suggestions of their form in Tunicates but not validated yet. In the third column, we show examples of diversification of specialized receptors acting in both phagocytosis and cytotoxicity. TLR = Toll-like receptors, TCR = T-cell receptor, MHC = Major histocompatibility complex, KIR = Killer Inhibitory Receptors, BCR = B cell receptor, VLR = Variable lymphocyte receptor, BHF = Botryllus histocompatibility factor, TRF = Transformer proteins, NLR = NOD like receptors, DSCAM = Down syndrome cell adhesion molecule, FREP = Fibrinogen-related proteins, NK-L = Natural killer-like receptor. ?—Represents missing functional or molecular validation. Created with BioRender.com. Data sources are summarized in Table 1.

**Table 1 cells-10-01853-t001:** Data sources for Figure 2.

Animal Group	Effector Cells	Receptors and Effector Molecules
Mammals	(Wolff and Humeniuk 2013) [105]	(Vandendriessche, Cambier et al., 2021) [19]
	(Rosental, Appel et al., 2012) [50]	
	(Hirayama, Iida et al., 2017) [20]	
	(Geffner 2005) [71]	
Osteichthyes	(Potts and Bowman 2017) [106]	(Wei, Zhou et al., 2007) [107]
	(Moss, Monette et al., 2009) [108]	(Kasahara and Flajnik 2019) [90]
	(Tang, Iyer et al., 2017) [109]	(Flajnik and Du Pasquier 2004) [75]
	(Flajnik 2018) [65]	
Agnathans	(Han, Das et al., 2015) [110]	(Pancer, Amemiya et al., 2004) [83]
	(Hirano, Guo et al., 2013) [111]	
	(Das, Li et al., 2015) [61]	
	(Mayer, Uinuk-ool et al., 2002) [62]	
Tunicates	(Rosental, Kowarsky et al., 2018) [16]	(Voskoboynik, Newman et al., 2013) [59]
	(Rosental, Raveh et al., 2020) [17]	(Nyholm, Passegue et al., 2006) [112]
		(Peronato, Franchi et al., 2020) [25]
Echinoderms	(Arizza, Giaramita et al., 2007) [113]	(Yakovenko, Donnyo et al., 2021) [32]
	(Cooper 1980) [48]	(Chou, Lun et al., 2018) [33]
	(Lin, Zhang et al., 2001) [56]	
Arthropods	(Muñoz-Chápuli, Carmona et al., 2005) [114]	(Melcarne, Lemaitre et al., 2019) [115]
	(Lanot, Zachary et al., 2001) [116]	
	(Meister and Lagueux 2003) [117]	
	(Cattenoz, Sakr et al., 2020) [15]	
	(Csordás, Gábor et al., 2021) [14]	
	(Cárdenas, Dankert et al., 2004) [78]	
Molluscs	(Franceschi, Cossarizza et al., 1991) [51]	(Schultz, Bu et al., 2018) [76]
	(Nakayama, Nomoto et al., 1997) [118]	(Loker, Adema et al., 2004) [77]
Sipunculids	(Boiledieu and Valembois 1977) [52]	(Flajnik and Du Pasquier 2004) [75]
Platyelminthes	(de Oliveira, Lopes et al., 2018) [119]	(Peiris, Hoyer et al., 2014) [28]
	(Morita 1991) [11]	
Cnidarians	(Snyder G 2021) [10]	
Porifera	(Mukherjee, Ray et al., 2015) [9]	(Pita, Hoeppner et al., 2018) [120]
		(Yuen, Bayes et al., 2014) [21]
		(Wiens, Korzhev et al., 2005) [22]

## Data Availability

Not applicable.

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
