# Peer review of "Evolution of Cellular Immunity Effector Cells; Perspective on Cytotoxic and Phagocytic Cellular Lineages"

_cells, 2021, doi:10.3390/cells10081853_

Round 1
Reviewer 1 Report
In their article, the authors summarize key aspects of phagocyte development and cytotoxicity. The work is relevant for several reasons and provides an excellent review of the literature cited. In the introduction, the evolution of phagocytosis is outlined.
Note 1: Here it seems useful to also distinguish from endocytosis in protozoa. Restricting it only to multicellular organisms is then useful in the article.
Phagocytosis is presented very clearly. Cells, receptors, effector molecules, and pathways are presented, and a good Figure 1 rounds out the section. A similar chapter follows for phagocytosis. This is followed by a presentation of the consequences.
Note 2: In my opinion, a critical discussion of the use of model organisms to study human cell function is missing here.
An excellent bibliography concludes the manuscript.
Author Response
Answer to reviewer 1:
We thank the reviewer for their positive opinion of the manuscript and the topic, and his/her insightful additions to the manuscript.
All changes are marked in yellow.
As suggested for Note 1, we emphasize that the focus of this manuscript is multicellular organisms with specialized immune effector cells. This includes: Abstract line 31, Introduction lines 71-72, and we emphasize on comparison between pinocytosis and phagocytosis in lines 103-104 of the phagocytosis effector cells part.
Thank you for Note 2, we agree that there was no discussion on the importance of studying model organisms' comparative immunology for human research and medical research. Accordingly, we have added a whole new paragraph to concluding remarks stating all the big discoveries coming from comparative immunology research and what was the source model organism for them. This includes examples we already talked about in the manuscript and two examples of anti-viral mechanisms. We think that this is now finishing the manuscript on a strong note, and emphasizing the importance of comparative immunology research. Thank you for it. Lines 482-496.
Reviewer 2 Report
The manuscript aimed at reviewing the current knowledge on the evolution of innate immunity . Through a comparative immunology approach the authors highlighted the difference and the similarity on functional commitment of phagocytic and cytotoxic cellular lineage at both cellular and molecular levels. The subject matter is of interest and manuscript well written, however some issues should be better addressed.
Specific comments:
In section 4 the interplay between inflammation and regeneration should be reported and discussed in detail as well as the fetal skin healing of mammals.
Author Response
Answer to reviewer 2:
We thank the reviewer for their positive opinion of the manuscript and the topic, and his/her insightful additions to the manuscript.
As suggested in section 4 we made a much more detailed discussion on the interplay between inflammation and regeneration. Additionally, we discuss in detail the fetal skin healing in mammals.
Added lines 437-466.
All changes are marked in yellow.